# Inhibition within a premotor circuit controls the timing of vocal turn-taking in zebra finches

Jonathan I. Benichov [1,2] & Daniela Vallentin [1,2]*

Vocal turn-taking is a fundamental organizing principle of human conversation but the neural circuit mechanisms that structure coordinated vocal interactions are unknown. The ability to exchange vocalizations in an alternating fashion is also exhibited by other species, including zebra finches. With a combination of behavioral testing, electrophysiological recordings, and pharmacological manipulations we demonstrate that activity within a cortical premotor nucleus orchestrates the timing of calls in socially interacting zebra finches. Within this circuit, local inhibition precedes premotor neuron activation associated with calling. Blocking inhibition results in faster vocal responses as well as an impaired ability to flexibly avoid overlapping with a partner. These results support a working model in which premotor inhibition regulates context-dependent timing of vocalizations and enables the precise interleaving of vocal signals during turn-taking.

[1] Institute of Animal Behavior, Freie Universität Berlin, Takustraße 6, 14195 Berlin, Germany. [2] Neural Circuits for Vocal Communication, Max Planck Institute for Ornithology, Eberhard-Gwinner-Straße, 82319 Seewiesen, Germany. *email: daniela.vallentin@orn.mpg.de

Spoken conversations often consist of rapid exchanges of vocalizations that are timed to minimize overlapping elements[1]. This form of vocal turn-taking involves the ability to precisely control the onsets of utterances and coordinate gaps. Individual speakers can respond to their conversational partners with varying response latencies[2] and the timing of vocal replies can depend on social context[3]. Although this behavior is well characterized we know little about the neural mechanisms that flexibly control when to initiate, delay, or withhold a response to a partner's vocalizations.

Other species also engage in vocal turn-taking[4]. Some mammals can produce antiphonal vocalizations[5–7] and context-dependent control of this behavior has been shown in several cases[6,8,9]. Many songbirds are notable specialists in vocal turn-taking as they can perform temporally precise song duets during the cooperative defense of territories or courtship displays[10–13]. These duetting behaviors are often highly complex sequences and involve the coordination of a variety of vocalizations. When attempting to identify the mechanisms specifically underlying the timing of interactions, it is advantageous to study a temporally coordinated vocal behavior with minimal acoustic complexity. Zebra finches, for instance, exchange thousands of ~50–100 ms long, flat harmonic 'stack' calls and slightly frequency-modulated harmonic 'tet' calls per day[14]. These call interactions are structured in a context-dependent manner[15,16] and serve an important role in group cohesion and pair bonding[15,17–19].

It has been shown that call-like vocalizations can be evoked by electrical stimulation of the midbrain area known as the dorsomedial part of the intercollicular nucleus (DM)[20] but more recent evidence suggests that the cortical forebrain pathway, necessary for the generation of learned courtship song[21], is involved in social calling as well[16,17,22,23]. In particular, the cortical premotor nucleus HVC (used as a proper name) is well situated to time vocalizations during turn-taking because it receives direct auditory inputs[24] and is known to guide the temporal pattern of song[25]. In addition, it has been shown that the downstream targets of HVC are necessary for coordinated call timing[16].

To ask how zebra finches adjust vocal timing during antiphonal calling we establish an interactive behavioral paradigm, exposing zebra finches to different social contexts and call playbacks. After demonstrating that the birds can flexibly adapt their call timing to social partners we explore the neural dynamics underlying vocal turn-taking. We then pharmacologically inactivate the nucleus HVC and establish its role in call timing. To further explore the contribution of single neurons to the generation of calls we carry out intracellular recordings of identified premotor neurons and inhibitory interneurons in HVC during vocal interactions. Both cell types display activity at the onset of call production, however, on average inhibition occurs before excitation. To test the hypothesis that interneurons are critically involved in delaying a vocalization in response to a vocal partner, we pharmacologically limit the influence of inhibition and detect accelerated call responses. These results support a working model in which inhibition regulates the initiation of vocal production during coordinated interactions.

## Results

**Zebra finches adapt call timing to avoid overlapping**. To characterize how zebra finches adjust their vocalizations during interactions, we set up a game of chicken—that is to say, a situation with a high potential for temporal conflict in which two birds are likely to call simultaneously. We first paired individual male zebra finches with an artificial partner (i.e., isochronous stack call playbacks at a rate of 1 Hz). In line with previous work,

we found that each bird responds to this predictable partner with a stereotyped latency (range: 198–332 ms, average response latency ± s.d. = 260 ± 39 ms, $n = 19$ birds, Fig. 1a–c, f). We then formed vocal triads consisting of pairs of latency-matched birds and the artificial partner (Fig. 1a–e). Given this more challenging context, birds could either call simultaneously as they respond to heard calls or they could coordinate their vocal timing to avoid overlapping. We found that in each triad, one or both birds adjusted their call response times to avoid overlapping (Fig. 1d–i), typically resulting in a three-call sequence starting with the call playback, followed by Bird 1 and then Bird 2. This occurred without practice sessions or prior pairing of birds. We calculated the differences in response latencies when responding to the playback alone or within a triad. In three out of four pairs we found that the timing of each bird's responses diverged when calling in a triad (Fig. 1g). The pair that did not exhibit a clear divergence in response time probabilities showed an alternative strategy to avoid overlapping. While their overall response latency distributions did not differentiate, both birds alternated their response sequence order across response cycles (e.g., playback, Bird 7, Bird 8 then playback, Bird 8, Bird 7) (Supplementary Fig. 1a, b).

The observed changes in response timing could have possibly resulted from one bird preferentially responding to the other bird rather than the playbacks, thereby obviating the need for controlled changes to vocal response timing. Alternatively, birds may anticipate the calls of a vocal partner and control their own call timing to avoid overlapping. In order to examine if the changes in call timing were simply reactive or whether they involved more adaptive control of call timing, we analyzed catch cycles. These consist of responses in which the typically later responding bird called first or alone (i.e., these calls were not direct responses to the other bird). During catch cycles, we also observed temporally shifted responses in all pairs except the pair with the alternative strategy (Fig. 1h).

To determine if birds in a triad overlap as often as expected if they maintained their response characteristics displayed while alone with playbacks (i.e., no behavioral flexibility), we performed a Monte Carlo simulation of the occurrences of call overlaps of latency-matched birds in the triad context. For this simulation, we used each bird's response rates and latency distributions, extracted from the alone with playback condition, as priors. We found that observed call overlap was significantly lower than predicted by the simulation (Fig. 1i). These findings demonstrate that call timing is flexible and dependent on social context. To test whether birds also change their call structure within different contexts we measured the acoustic structure of calls produced alone with the artificial partner or in a triad. Zebra finches neither altered the duration nor the spectral features of calls across contexts (Fig. 1j; Supplementary Fig. 1c, d) indicating that vocal timing can be controlled independently from acoustic structure during interactions.

**The premotor nucleus HVC regulates precise call timing**. The premotor nucleus HVC exhibits stereotyped activity during singing[23] and auditory-evoked activity in response to the tutor's song[26]. Due to HVC's role in patterned vocal output and auditory processing, this nucleus might also be involved in vocal turn-taking. To test the hypothesis that HVC is necessary for the regulation of call timing we inactivated HVC with bilateral infusion of muscimol, a GABA$_A$ agonist, and measured birds' responses to call playbacks (Fig. 2a–c). We found that blocking HVC's influence did not prevent birds from calling but reduced call response rates on average (Fig. 2d). Notably, inactivation reversibly impaired the precision of response timing compared to

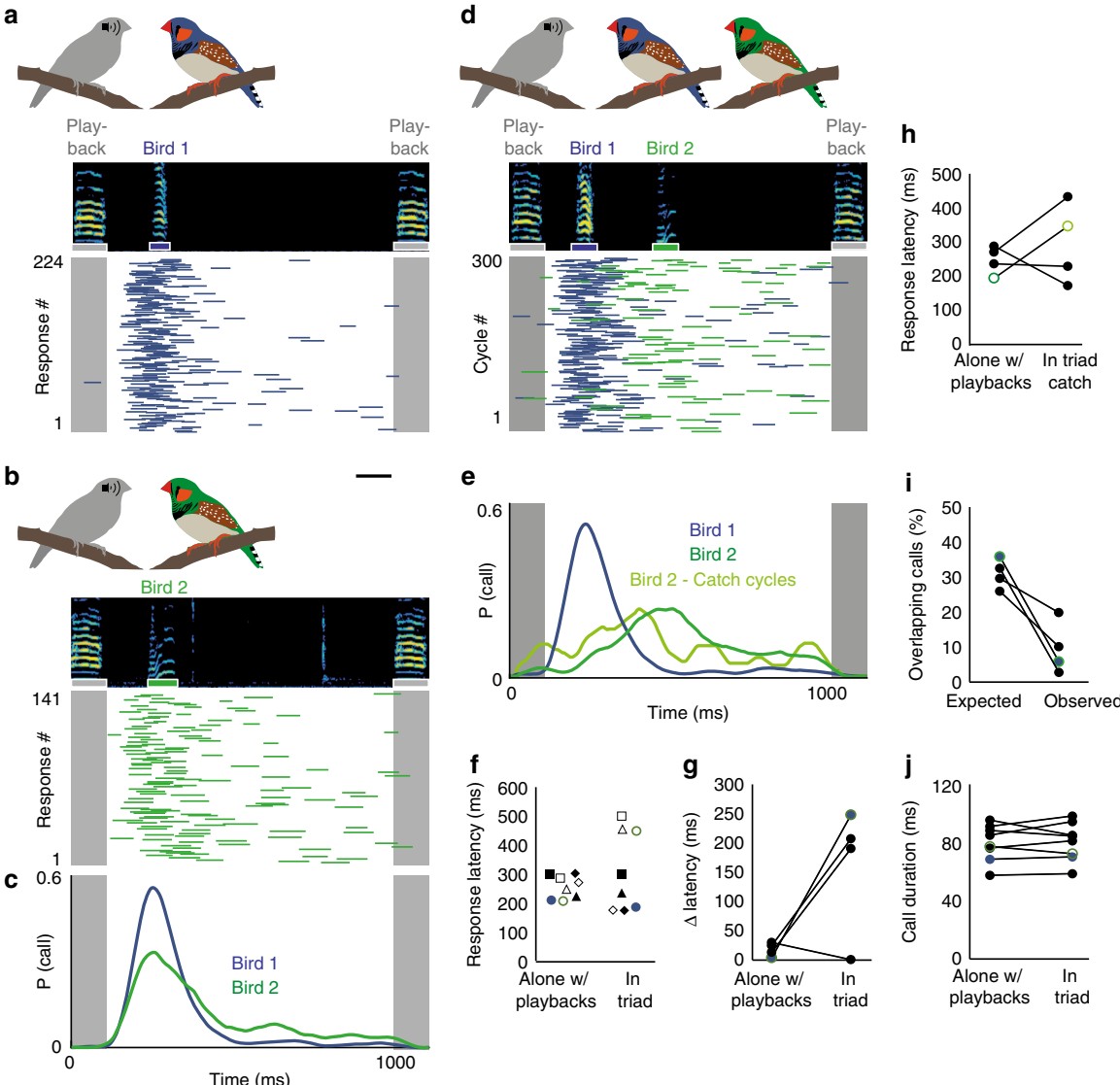

**Fig. 1 Call coordination in zebra finches. a**, **b** Call responses to call playbacks presented once per second for a 10-minute session while Bird 1 (**a**) and Bird 2 (**b**) were each interacting with the playbacks alone. (*top*) Spectrogram of call playback and bird's call response. (*bottom*) Call playback indicated in gray. Bird 1 or 2 call responses indicated by blue and green bars, respectively. Scale bar, 100 ms. **c** Call response probability distributions for Bird 1 (blue) and Bird 2 (green) with matched peak response latencies (202 ms, 198 ms). **d** Call responses of Bird 1 (blue) and 2 (green) when housed together and presented with call playbacks (in triad). **e** Response distributions for birds in (**d**) when calling in triad. Light green distribution represents catch cycles in which a call from Bird 1 did not precede a call from Bird 2. **f** Mean response latencies for 8 birds alone with call playbacks (coefficient of variation CV (response latencies across birds) = 0.16) vs. in triad with a latency-matched bird and playbacks (CV (response latencies across birds) = 0.45). Same shapes indicate matched pairs. Blue and green markers represent example pair from (**a–e**). **g** Difference (Δ) in response latencies between matched birds for 4 pairs when alone with playbacks vs. in triad (mean alone w/ playbacks ± standard deviation = 18 ± 12 ms, mean in triad = 165 ± 112 ms). The pair that did not change its latency developed an alternating strategy (see Supplementary Fig. 1). **h** Call response latencies (alone with playbacks vs. triad catch cycles) for those birds in each pair that had a greater shift in call timing. **i** Expected vs. Observed percent of overlapping calls for all 4 latency-matched pairs (expected overlap = 30.9% ± 4.3%, Observed overlap = 9.2% ± 7.6%, Wilcoxon rank-sum test, $p = 0.029$, $n = 4$ pairs). **j** Duration of calls when alone with playbacks or when in triad (mean duration alone w/ playbacks = 81 ± 13 ms, mean duration in triad = 81 ± 13 ms, Wilcoxon sign rank test, $p = 0.779$, $n = 8$ birds).

controls (Fig. 2a–c, e, f). This temporary loss of precision reproduces the effect of lesioning the downstream motor area, the Robust nucleus of the Archopallium (RA)[16], suggesting that HVC is the primary site of call timing regulation. Upon closely examining the acoustic features of calls, we found that the pitch and spectral structure of calls changed during treatment with muscimol, relative to saline control (Fig. 2g–j; Supplementary Fig. 2). This suggests that HVC may also influence the acoustic structure of short calls in addition to its role in the control of timing.

**Inhibition precedes premotor activity in HVC during calling.** We then asked how the neural activity within HVC controls call timing. Therefore, we performed intracellular recordings of antidromically identified HVC-RA projecting premotor neurons during vocal interactions by using a motorized microdrive[27,28]. We identified premotor neurons (14/16 neurons) that exhibited bursts of action potentials corresponding to the onset of short and acoustically simple tet and/or stack calls (burst onset time = −10 ± 22 ms relative to call onset, $n = 5$ neurons exhibited spikes with stack call and 11 neurons with tet calls, 2 neurons with both,

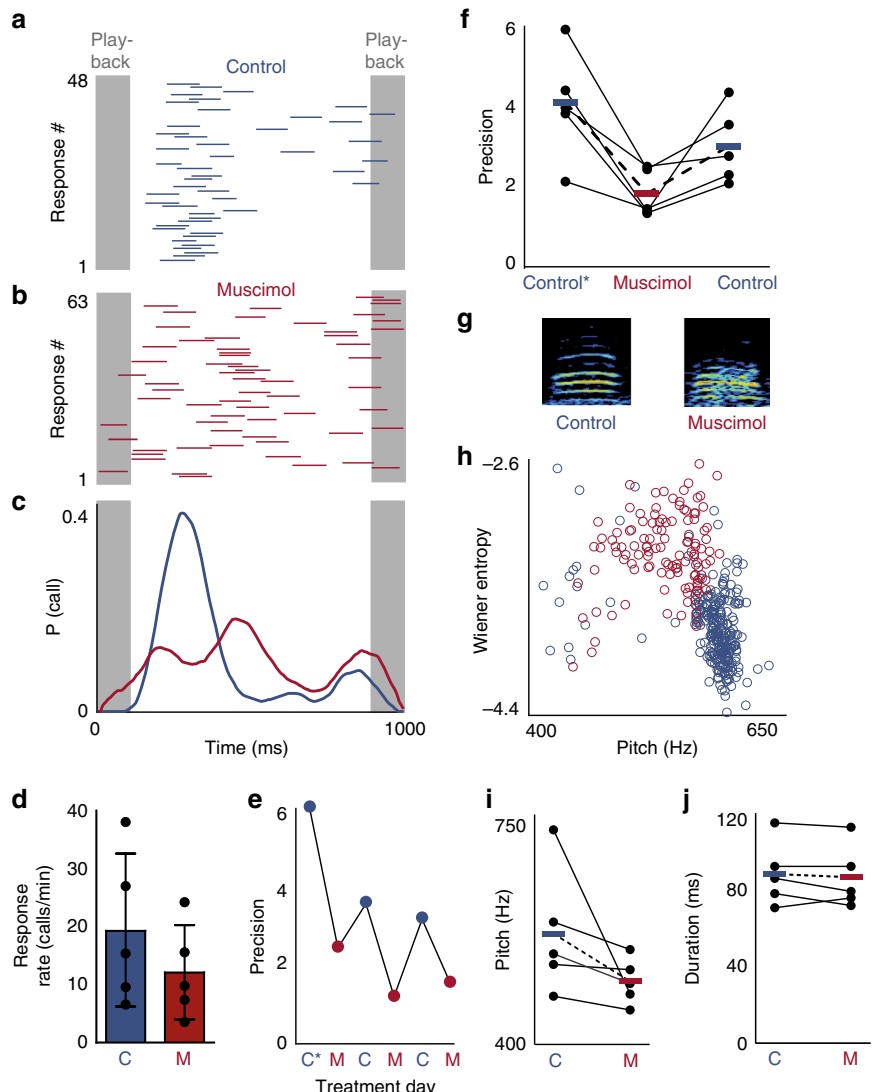

**Fig. 2 HVC is required for precise timing of calling behavior. a** Call responses (blue) to call playbacks (gray) for example bird during infusion of saline.
**b** Call responses during bilateral infusion of 5 mM muscimol. **c** Response probability distributions for control (blue) and muscimol (red) conditions in (**a, b**).
**d** Average response rate during control and muscimol conditions (control = 19.32 ± 11.66 calls/min, muscimol = 12.12 ± 7.19 calls/min, Wilcoxon signed-
rank test, $p < 0.05$, $n = 5$ birds), error bars: standard deviation. **e** Effects of HVC inactivation on response latency precision for an example bird across
6 days starting with pre-surgery (C*) and alternating daily between muscimol (M) and saline infusion (C). **f** Response latency precision assessed pre-
surgery (control*), during HVC inactivation (muscimol), and during saline (control) infusion (mean precision: C* = 4.13 ± 1.37, M = 1.83 ± 0.60, C = 3.02 ±
0.95, one-tailed Wilcoxon signed-rank test, C* vs. M: $p = 0.031$, C vs. M: $p = 0.031$, $n = 5$ birds, bars and dotted lines represent means). **g** Spectrograms
of two calls recorded during a control condition (left) and a muscimol infusion (right). **h** Effects of HVC inactivation on acoustic features of calls for
example bird in (**g**). Wiener Entropy i.e. the noisiness of calls (where white noise would have a value of 0 and a pure tone would have a large negative
value) increases when muscimol is applied (red circles, $n = 121$ calls) whereas the pitch is higher in the saline control condition (blue circles, $n = 261$ calls).
**i** Median pitch during saline control (blue) and muscimol (red) application (mean pitch ± s.d. for control = 568.5 ± 97.4 Hz, pitch for muscimol = 495.6 ±
35.1 Hz, $n = 5$ birds). **j** Duration of calls during control and muscimol conditions (mean duration for control = 89 ± 17 ms, mean duration for muscimol =
87 ± 17 ms, Wilcoxon signed-rank test, $p = 0.3750$, $n = 5$ birds). Source data is available as a Source Data file.

in 10 birds, Fig. 3a–c), potentially serving as a go signal for these
vocalizations as they are exchanged during vocal turn-taking.

Because of HVC's critical role in song production[23,28] and the
possibility of tet- and stack-like elements being incorporated into
songs during vocal development[29], we tested whether individual
premotor neurons can be involved in both song and call
production. To do so, we recorded from 10 neurons in 5 birds
during both behaviors. We found that 6/10 HVC premotor
neurons generated bursts of action potentials at and before call
onsets as well as during song production (Fig. 3c–e). Overall,
the spiking profiles differed during calls and songs (Fig. 3e).
Because these neurons were active during both vocal contexts we

wondered if specific acoustic features followed the activity of
particular neurons. To test this, we measured the pitch and
harmonic structure (represented as Goodness of Pitch[30]) for calls
and song elements following spiking onset (Fig. 3f, g). We did not
detect a correlation between vocalization types in either measure,
suggesting that these neurons function in networks that generate
multiple motor patterns.

In addition to the premotor neurons that spiked at call onset, we
observed 5/16 HVC premotor neurons that were actively
suppressed prior to at least one call type (hyperpolarization =
−6.85 ± 1.94 mV). Of these neurons that were also recorded during
singing ($n = 4$ neurons), all cells exhibited canonical stereotyped

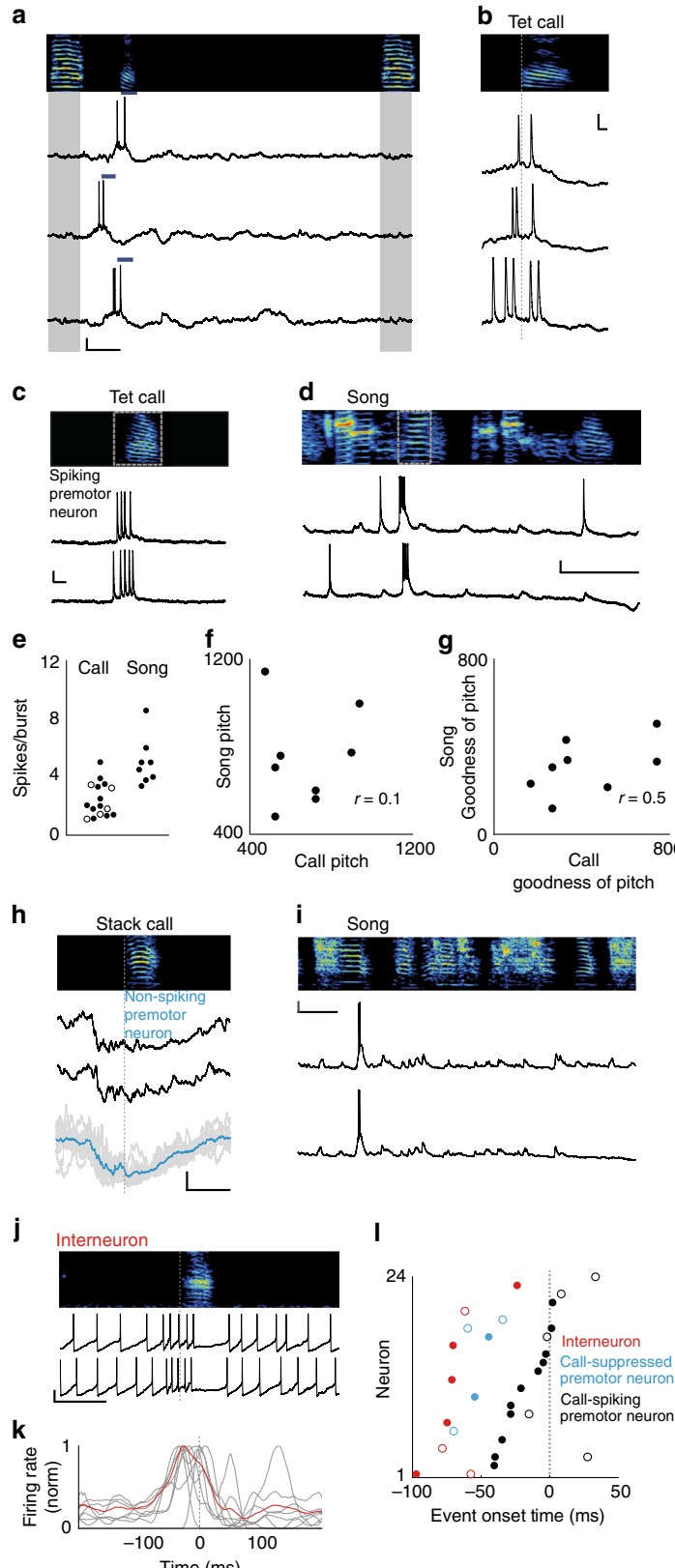

bursts during song (Fig. 3h, i). Upon comparing the onset of activity in spiking neurons to the onset of hyperpolarization in suppressed neurons, we found that call-related inhibition preceded call-related premotor activity (Fig. 3h, l).

This pattern of early inhibition led us to examine the activity of inhibitory interneurons within HVC, which are densely interconnected with premotor neurons[31,32]. By recording from HVC interneurons during calling, we found that these cells ($n = 7$ neurons, in 6 birds) showed a transient increase in firing rate associated with calling, followed by a reduction in their firing rate (Fig. 3j, k; Supplementary Fig. 3). The increase in call-related interneuron activity precedes that of call-related premotor activity

**Fig. 3 Activity of HVC neurons preceding call production. a** Spectrogram (top) and intracellular recordings of an HVC premotor neuron during call responses (blue bars) to call playbacks (gray). *x, y* scale bars: 100 ms, 10 mV. **b** Intracellular activity as in (**a**), aligned to tet call onsets (gray dotted line). *x, y* scale bars: 10 ms, 10 mV. **c, d** Bursting activity from a premotor neuron for tet calls (**c**) and song (**d**). *x, y* scale bars for (**c**): 10 ms, 10 mV. *x, y* scale bars for (**d**): 100 ms, 10 mV **e** Mean spikes per burst for premotor neurons during calls and song (*n* = 14 neurons (calls), *n* = 6 neurons (song), 2.4 ± 1.2 spikes per burst (calls), 5.0 ± 1.7 spikes per burst (songs), Wilcoxon rank-sum test, *p* < 0.001, Note: two neurons elicited two bursts during singing). Tet calls: solid circles, stack calls: open circles. **f, g** Correlation of acoustic features occurring after premotor neuron activity (gray boxes in (**c, d**)), (Spearman correlation, pitch: *p* = 0.840, goodness of pitch: *p* = 0.197, *n* = 6 neurons). **h** Hyperpolarization of a premotor neuron prior to calls (black). Bottom: Overlay of 13 renditions from example neuron in gray. Mean membrane potential in blue. *x, y* scale bars: 100 ms, 5 mV. **i** Same premotor neuron as in (**h**) recorded during two renditions of song. *x, y* scale bars: 100 ms, 10 mV. **j** Example recording of interneuron during calling. **k** Normalized firing rates for 7 interneurons relative to call onsets (average indicate in red). *x, y* scale bars: 100 ms, 10 mV. **l** Red circles: Onset of increased activity for *n* = 7 interneurons in (**k**), onsets of hyperpolarization for *n* = 5 premotor neurons (blue circles), and onsets of bursts for *n* = 14 premotor neurons (black circles) relative to start of call (gray dotted line). Open circles: stack calls, closed circles: tet calls (mean premotor neuron burst onset = −10 ± 22 ms, mean hyperpolarization onset = −52 ± 14 ms, Wilcoxon rank-sum test, *p* = 0.003, mean interneuron firing increase onset = −56 ± 31 ms, Kruskal–Wallis test, *p* < 0.0001). Source data is available as a Source Data file.

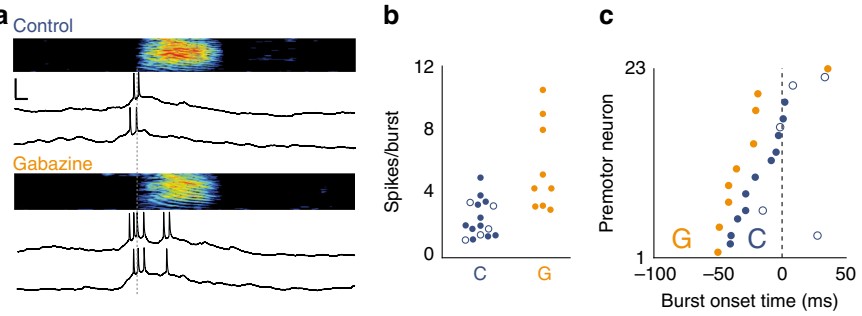

**Fig. 4 Effects of local gabazine microinfusion on HVC premotor neuron spiking during call production. a** Top: Example of call-related bursting for a premotor neuron recorded during saline control. Bottom: Same cell as (top) but recorded during local gabazine infusion. *x, y* scale bars: 10 ms, 10 mV. **b** Number of call-related spikes per burst in premotor neurons recorded under control or gabazine conditions (control: 2.4 ± 1.2 spikes per burst, *n* = 14 neurons in 10 birds, 2 were active during both call types; gabazine: 5.7 ± 2.8 spikes per burst, Wilcoxon rank-sum test, *p* = 0.003, *n* = 9 neurons in 3 birds). **c** Mean burst onset latency relative to call onsets for premotor neurons recorded during control conditions (blue) vs. premotor neurons recorded during gabazine microinfusion (orange), (spiking onset for control: −10 ± 22 ms, *n* = 14 neurons, for gabazine: −26 ± 26 ms, Wilcoxon rank-sum test, *p* = 0.039, *n* = 9 neurons). Source data is available as a Source Data file.

but does not differ from the onset of hyperpolarization in call-suppressed premotor neurons (Fig. 3l). This temporal relationship suggests that inhibitory interneurons play a primary role in specifying if premotor neurons are active during calling. Furthermore, this sequence of activity may also influence when call-spiking premotor neurons are active, thereby regulating call timing.

**Disinhibition of HVC increases call-related premotor drive.** We tested whether inhibition within HVC affects the activity of call-spiking premotor neurons by recording intracellularly from premotor neurons during call production while focally applying the GABA_A antagonist gabazine (Fig. 4). This application of gabazine, which limits GABAergic inhibition of premotor neurons, was restricted to a small region of HVC and had no effect on call production. However, premotor drive was significantly higher in terms of spikes per burst when gabazine was applied (Fig. 4a, b). With respect to timing, we observed that the first action potential occurred earlier relative to call onset compared to saline control conditions (Fig. 4c). Thus, inhibition likely plays an important role in shaping descending premotor activity and thereby coordinates the initiation of a call. As expected, none of the recorded premotor neurons were hyperpolarized prior to call onset under gabazine treatment. Together, these results suggest that inhibitory interneurons limit spiking activity to a restricted group of premotor neurons and mediate those premotor neurons that do trigger call production.

**Disinhibition of HVC leads to changes in call timing.** The modulation of premotor neuron activity by local inhibition led us to investigate the effects of HVC disinhibition on calling behavior. If these physiological changes reflect a causal role of inhibition in regulating calling behavior, we should expect the reduction of inhibition in HVC to affect call timing. To test this hypothesis, we disinhibited HVC with bilateral infusion of gabazine and paired birds with the call playbacks. There was no consistent effect on call response rate when gabazine was infused (Fig. 5j). However, we found that the lack of inhibition within HVC results in significantly faster and, in some cases, less variable call response timing relative to saline control infusions (Fig. 5a–e, Supplementary Movie 1). This manipulation also increased the variability of the acoustic structure of calls for four out of five birds (Fig. 5f–i; Supplementary Fig. 4) potentially due to an increase in number of active premotor neurons. The effect on timing indicates that auditory stimulation with calls can more rapidly trigger vocal responses by activating premotor circuitry that has been released from inhibition. Together, these results show that blocking the impact of inhibition within HVC accelerates premotor drive and diminishes a bird's ability to delay the timing of their vocalizations relative to a vocal partner.

To study the impact of inhibition on adaptive call coordination, we tested birds with a jamming avoidance paradigm[16]. In contrast to the effect of muscimol, gabazine application preserved response latency stereotypy, allowing us to identify a latency window of high calling probability in response to the call playbacks. We then inserted an additional playback, a so-called jamming call, when the bird was most likely to respond. During

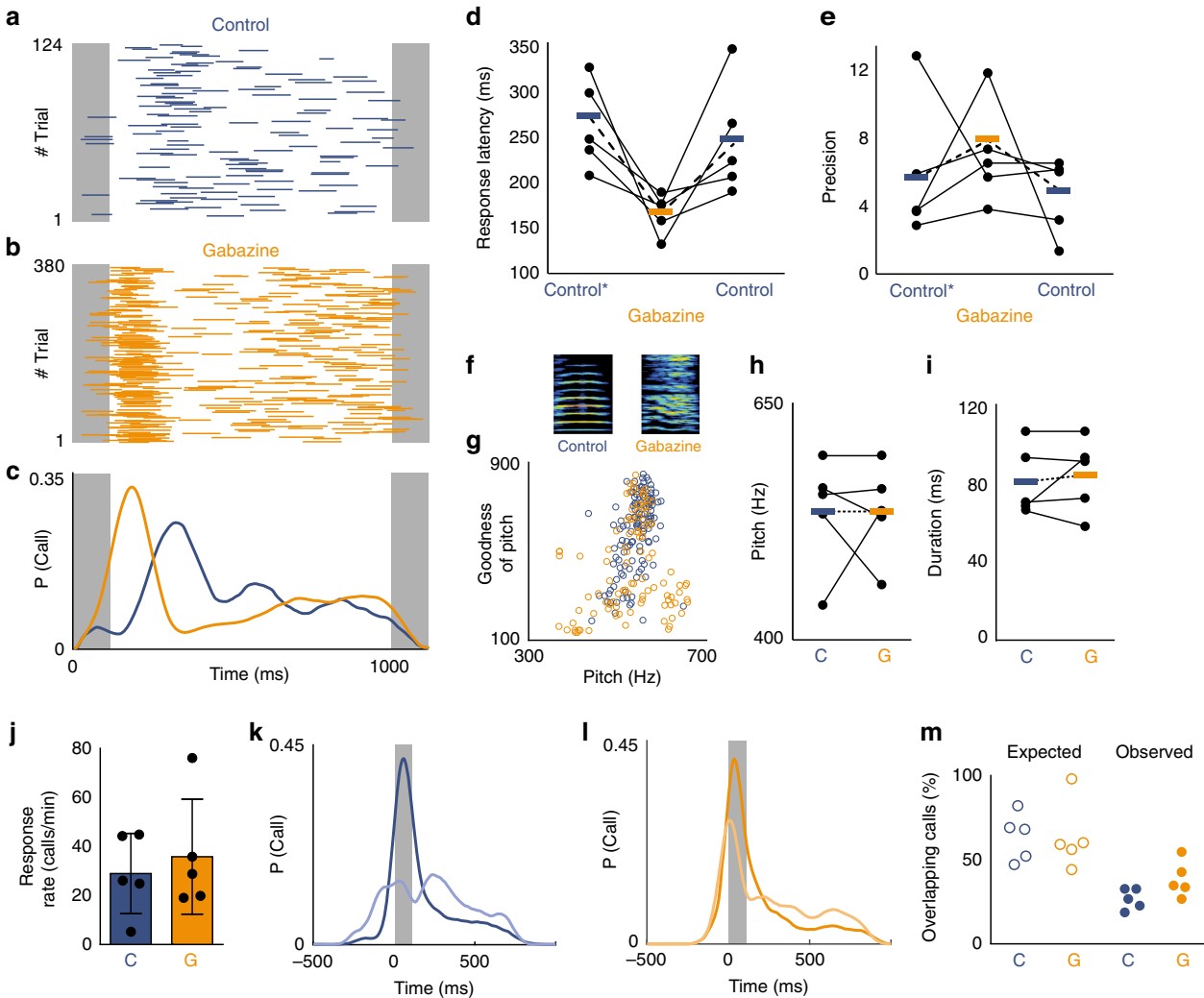

**Fig. 5 Inhibition within HVC regulates call timing. a** Call responses (blue bars) to call playbacks (gray) during bilateral infusion of saline in HVC (control). **b** Accelerated call responses (orange bars) during infusion of 0.01 mM gabazine. **c** Response probability distributions for control (blue) and gabazine (orange) conditions in (**a**, **b**). **d** Call response latency significantly decreases during the gabazine condition (orange) compared to pre-surgery (control*) and saline control conditions (blue). Response latency: control* = 266 ± 51 ms, gabazine = 166 ± 25 ms, control = 249 ± 68 ms, Kruskal–Wallis test, $p = 0.0081$, $n = 5$ birds). **e** Mean Precision: control* = 5.64 ± 4.25, gabazine = 7.73 ± 3.46, control = 4.42 ± 2.36, Kruskal–Wallis test, $p = 0.164$, $n = 5$ birds). **f** Spectrograms of example calls from one bird during the control (left) and gabazine (right) conditions. **g** Effects of reduced inhibition on call acoustic features for example bird shown in (**f**). **h** Call pitch during control and gabazine condition (mean pitch (control) = 530 ± 59 Hz, mean pitch (gabazine) = 530 ± 50 Hz, Wilcoxon signed-rank test, $p = 1$, $n = 5$ birds). **i** Call duration during control and gabazine conditions (mean duration control: 81 ± 18 ms, gabazine: 84 ± 19 ms, Wilcoxon sign rank test, $p = 1$, $n = 5$ birds). **j** Response rates during control vs. gabazine infusions (mean control: 29.0 ± 16.4 calls/min, gabazine: 35.9 ± 23.5 calls/min, Wilcoxon sign rank test, $p = 1$, $n = 5$ birds). Error bars: standard deviation. **k. l** Call response distributions for jamming avoidance task during (**k**) control (blue) and (**l**) gabazine (orange) infusions. Gray box: normalized jamming window, dark: baseline, light: during jamming call playbacks ($n = 5$ birds). **m** Percent of calls expected to overlap with the jamming window based on response to 1 Hz call playbacks (left). Percent of calls overlapping with jamming playbacks (right) (mean control: 25 ± 6% of calls vs. gabazine: 37 ± 11% of calls overlapping with jamming playback, Wilcoxon rank-sum test, $p = 0.0459$, $n = 5$ birds). Source data is available as a Source Data file.

control conditions, all 5 birds overlapped with the jamming call at rates lower than expected based on their response times to 1 Hz calls. However, during gabazine treatment, these birds failed to reduce their rates of overlap to levels observed during the control condition (Fig. 5k–m). In summary, the flexible timing of calls in response to different contexts depends on an intact inhibitory-excitatory interplay within the premotor circuit HVC.

## Discussion

In this study, we examined the neural mechanisms underlying vocal turn-taking in zebra finches and determined that inhibition in the cortical premotor nucleus HVC provides a critical

mechanism for regulating interactive vocal timing. Although adult male zebra finches have a limited ability to adjust spectral features of songs and calls, the flexibility in the timing of their vocalizations appears to provide a means by which more complex patterns of communication can be achieved. This behavior might serve as an important tool for maintaining specific lines of communication in social groups. For instance, flexible modulation of call timing in response to different vocal partners and contexts is a potential strategy for maintaining and updating social networks and coding for individual identity[33,34]. Because zebra finches live in large groups, acoustic interference is a common challenge they need to overcome. One strategy is to vocalize louder in a noisy environment[35,36]. In this study, we

observed that zebra finches can also adjust the timing of their calls relative to a partner, which represents an alternative strategy to cope with acoustic masking. A similar principle has been observed in frogs[37], insects[38] as well as in mammals[39]. We observed that birds adjust their response latency and call order within groups without extensive reinforcement or practice. How groups of birds converge on strategies is an intriguing direction for further ethological investigation and likely involves additional social factors[40]. Visual cues might also play a role in structuring collective vocal sequences[41].

Previous studies have shown that HVC premotor neurons control song timing[23,25,42]. We show that individual premotor neurons in HVC serve multiple functions; namely these sparsely firing neurons can be active during both call and song production. Although male zebra finches can call with bilateral HVC inactivation, presumably controlled by midbrain structures[20], we show that HVC is necessary for call timing precision. This may also be the case in female zebra finches, who do not sing, but can actively coordinate their calls and may rely on a reduced form of the vocal motor pathway to control their timing[16,40,43,44]. Our findings suggest that the cortical vocal-motor pathway impinges upon subcortical areas responsible for call production in order to control the timing of vocal output. In this framework, premotor neurons provide specificity and precision to vocal onsets whereas the premotor inhibition ensures that the initiation of vocalizations occurs at appropriate times. There is increasing evidence that this form of cortical control over subcortical vocal production centers is a shared feature in birds and mammals[6,45–47].

In humans, the capacity for vocal turn-taking emerges well before the first imitative utterance[48] and can be affected in Down syndrome[49], in premature infants[50], and in autism spectrum disorder[51]. Autism has been associated with an imbalance of excitation and inhibition where synaptic inhibition is decreased[52–54]. Identifying the source that informs inhibitory interneuron activity within premotor circuitry will lead to a better understanding of how precisely timed vocal turn-taking is achieved and, thus, might aid in developing strategies for clinical interventions in patients with impairments to social vocal coordination.

## Methods

**Animals.** Animal care and experimental procedures were performed with the ethical approval of the Landesamt für Gesundheit und Soziales (LAGeSo Berlin) at the Freie Universität Berlin and/or the Institutional Animal Care and Use Committee at the New York University Medical Center (NYUMC). For behavioral and electrophysiological experiments, adult male zebra finches (>90 days post hatching) were acquired from the breeding facility at the Freie Universität Berlin or obtained from an outside breeder for experiments conducted at the NYUMC.

**Call playbacks.** Call audio files were composed of natural calls recorded at 44.1 kHz sampling rate from an interacting pair-bonded male in a sound-attenuated chamber. These calls were representative of an average stack call and reliably elicited call responses in male birds. A 10 kHz pure tone marker (outside of audible range of zebra finches[55]) of the same duration and root mean square amplitude, was added to the call for identifying onsets/offsets in case of overlap. Calls were delivered at 70 dB through a speaker placed behind a mirror within the sound attenuated testing chamber. Call patterns generated were isochronous (rate of 1 call/s for ten 30 s blocks interspersed with 30 s intervals of silence) or consisted of jamming call pairs (one jamming pair per second) (Fig. 4k–m)[16].

**Call response recordings and analysis.** Responses were automatically segmented with Sound Analysis Pro (SAP 2011[30]) and manually segmented in case of overlap. Since tet and stack calls are used within the same affiliative behavioral contexts[56] we did not differentiate between the two in our response time analysis. Call response onsets and offsets were coded relative to the onset of the previous call playback. These onsets and durations were summed across all cycles in a session to produce a response probability distribution and smoothed with a moving average of 99 ms in 1 ms steps. Coded responses were used to generate raster plots and probability distributions in MATLAB (MathWorks, Inc., Natick, MA). Response latency was determined as the onset of the 100 ms window containing the highest

percentage of calling activity across all 1000 ms cycles. This measure was also applied for defining the jamming window during the jamming avoidance task.

Prior to testing, all birds had been housed in a common aviary. Birds were first placed in the testing sound box individually and presented with call playbacks. Birds that did not readily and reliably respond to playbacks were excluded from the experiment. Birds in triads were recorded in the same cage, separated by a visually and acoustically transparent divider with one of two matched-pair cardioid condenser microphones (Behringer C2) in each compartment. The amplitude differences between microphones were used to determine the identity of the caller. For the Δ Latency measure the response latencies for both individuals in each pair were subtracted. Catch cycles occurred when the bird with the greater change in average latency called alone or first in response to the call playback.

To estimate the expected call overlap of latency matched pairs, we performed a Monte Carlo simulation. We calculated the response rate and the timing of observed calls in the alone with playbacks condition. For each bird in a pair these data were randomly sampled 300 times (30 calls × 10 blocks) replicating an experimental session of the vocal triad. The percent of overlapping calls was calculated across trials. This procedure was repeated 1000 times and the average of the resulting distribution was reported as expected overlap in %. Code will be made available upon request. Acoustic features (pitch, goodness of pitch, Wiener entropy, & duration) were calculated for segmented calls using SAP 2011.

**Precision score.** Precision score is a measure of how reliably a call occurred in the same time window (100 ms) across renditions (e.g., 100 calls with exactly the same onset would give rise to a precision value of 12, whereas 100 calls with random onsets would have a precision value approaching 0). For each call in a session, the response onset latency differences to all other calls were measured. Then we calculated the proportion of these differences that were within ±50 ms (approx. duration of a call). These proportions were used to compute a Z-score, relative to a distribution of proportions from 1000 simulated sessions containing an equal number of uniformly distributed pseudorandom latencies. The precision score is expressed as the square root of the absolute value of the Z-score.

**Song analysis.** Acoustic features (pitch, goodness of pitch, Wiener entropy, and duration) were calculated using SAP 2011. The song segments analyzed began at song-related burst onsets and had a duration equivalent to the average time from call-related burst onset to call offset for each cell.

**Surgery.** In preparation for pharmacological experiments, zebra finches were first anesthetized with isoflurane (1–3% in oxygen). The center of HVC was located based on stereotactic coordinates (0.2 mm anterior, 2.3 mm lateral of the bifurcation of the midsagittal sinus) and bilateral, rectangular craniotomies (dimensions: 1.2 mm × 0.7 mm) were cut such that the lateral/anterior ends were oriented ~45 degrees away from the midline (Supplementary Fig. 2b). Until experiments were conducted, the craniotomies were protected using a silicone elastomer (Kwik-Cast; WPI). A custom-made stainless steel head plate was affixed to the skull using dental acrylic (Paladur, Kulzer International).

For electrophysiological recordings, we implanted the motorized intracellular microdrive. The zebra finch was first anesthetized with isoflurane (1–3% in oxygen). The base of the microdrive was then affixed to the skull of the bird using dental acrylic. For antidromic identification of HVC-RA projecting premotor neurons[23], a bipolar stimulating electrode was implanted into the downstream nucleus RA. After 1–4 days, we prepared a 100–200 µm diameter craniotomy above HVC and carefully removed overlying dura. A well was built around the craniotomy using silicone elastomer. To protect the brain from desiccation, the well was filled with either saline or a silicone gel (Dow Corning; 10,000 cSt) during behavioral and electrophysiological recordings and with a layer of silicone elastomer overnight.

**Pharmacological perturbations.** For HVC inactivation, the GABA$_A$ receptor agonist muscimol (5 mM muscimol in saline, warmed to 40 °C to approximate the body temperature of zebra finches) was applied bilaterally via saturated gel foam sponges (Avitene Ultrafoam, Bard) to the dorsal surface of HVC in head-fixed adult zebra finches[32]. Muscimol infusions have been shown to diffuse 0.5–1.0 mm through cortical tissue (approximately corresponding to the maximum depth of HVC), with immediate suppression of excitatory transmission upon contact with 10 µM solution[57]. Due to the presence of APH (area parahippocampalis) above HVC, a thin layer of this tissue (~10–100 µm thick) was resected along with dura mater using a fine tungsten pick, directly exposing the central dorsal portion of HVC. In an effort to restrict the site of pharmacological action to HVC, silicone elastomer wells were created around the perimeter of the craniotomies. Immediately following the application of the saturated gelfoam to the surface of HVC, the well was sealed over with silicone elastomer and the bird was released into the recording chamber (Supplementary Fig. 2a). After a 10 min period, behavioral testing proceeded as described above. For the saline condition, we followed the same protocol. We alternated saline controls and drug application on a daily basis. Before and after all experiments, craniotomies were cleaned of any overlying tissue and flushed with saline and fresh silicone elastomer was applied to seal the craniotomies.

For blocking inhibition within HVC in a set of different birds, the GABA$_A$ antagonist gabazine (0.01 mM) was applied bilaterally and the same protocol as described above was followed. Intracellular recordings during gabazine infusion were achieved with a small cannula positioned near the craniotomy after implantation of the intracellular microdrive. While the bird was socially interacting, 0.01–0.1 mM gabazine was applied directly to the surface of HVC and recordings were obtained as described below. Efficacy of surface infusion was confirmed with electrophysiological recordings[58,59] in which the effects of gabazine (increased number of spikes per burst and higher amplitude subthreshold activity) were observed in cells at depths down to 580 μm from the HVC surface. With these recordings, we also determined the time course of action to begin within less than 10 min of surface infusion.

**Electrophysiological recordings.** For intracellular recordings, sharp electrodes with an impedance of 70–130 mΩ were prepared using a modified horizontal micropipette puller (P-97; Sutter Instrument Company) and backfilled with 3 M potassium acetate. Zebra finches were briefly head fixed (without anesthesia) and partially immobilized in a foam restraint to allow for freshly prepared pipettes to be inserted into the microdrive. Once these electrodes were lowered into the brain and began to encounter spiking activity, the bird was released and intracellular recordings were pursued by lowering the pipette through HVC in ~5 μm steps. A brief (10–20 ms) buzz pulse was used to penetrate the membrane. Once a stable recording (spike height: > 40 mV, resting membrane potential: < −50 mV, recording duration: >3 min) was achieved, call playbacks or a female bird were presented to elicit calls and song. All membrane potential measurements were digitized at 40 kHz using a National Instruments acquisition board and acquired with custom MATLAB software.

In order to identify cell types, we stimulated RA with single biphasic (200 μs per phase) current pulses of <250 μA. HVC-RA-projecting premotor neurons were identified by responding to each pulse with a reliable antidromic spike with minimal latency jitter (<50 μs)[23]. For those cells recorded during singing, HVC-RA neurons were further confirmed by the following criteria: (1) song-related depolarization (2) 0–2 bursts of action potentials per motif (3) highly stereotyped subthreshold activity across song repetitions[27,28].

Interneurons were identified when they fulfilled at least 2 of the following criteria :(1) Depolarization during song, (2) displaying phasic spiking activity which is interrupted by short silent gaps and is stereotyped across song repetitions (Note: 6 of 7 interneurons were also recorded during singing and displayed the structured firing, with local minima in their spike rates[32,60]), (3) by their high firing characteristics and their high spike time jitter after antidromic stimulation, which was often accompanied by multiple spikes[23], (4) undershooting action potentials below resting membrane potential.

**Data analysis.** We used MATLAB for data analysis. If not noted differently, data are presented as mean ± standard deviation. The voltage traces recorded from each cell were aligned to the onset of each call or song rendition. Spikes were extracted using a thresholding algorithm. The time point of the first spike of a burst during call-related activity was taken as the spiking onset time of premotor neurons.

To determine the hyperpolarization of non-spiking HVC premotor neurons a baseline resting membrane potential during silence was calculated. The onset time of hyperpolarization was detected as the falling inflection point at which the mean subthreshold activity across call renditions deviated from the baseline. The hyperpolarization was then calculated by subtracting the voltage at the onset of hyperpolarization from the minimum voltage during a period between −100 ms to 100 ms from call onset. The onset time of firing rate change in interneurons is defined as the time point relative to call onset at which firing rate increased above two standard deviations of the mean baseline firing rate.

**Reporting summary.** Further information on research design is available in the Nature Research Reporting Summary linked to this article.

## Data availability
The data that support the findings of this study are available from the corresponding author upon reasonable request. Source data underlying Figs. 2d, 3e, f, g, 4b, 5j is provided as a Source Data file.

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

## Acknowledgements
We thank Constance Scharff and Michael Long for their support and help with the paper. We also thank Philipp Norton, Daniel Okobi, Linda Bistere, Fabian Heim, Susanne Seltmann, and Georg Kosche for reading and commenting on a previous version of the paper. The study was supported by an Emmy Noether grant (Project number VA742/2-1) funded by Deutsche Forschungsgemeinschaft (DFG, German Research Foundation) to D.V., an European Starting grant (ERC-2017-StG - 757459 MIDNIGHT) to D.V., a CRC grant by DFG – Project number 327654276 – SFB 1315 to D.V., and the Konishi Neuroethology Research award from the International Society of Neuroethology to J.B.

## Author contributions
J.B. and D.V. designed the research, performed experiments, analyzed data, and wrote the paper.

## Competing interests
The authors declare no competing interests.
