## [Peer Review File · Nature Communications]

Reviewers' comments:

Reviewer #1 (Remarks to the Author):

Vocal interactions among individuals involve vocal turn-taking. This type of social interaction is common even among animals that do not learn their vocalizations. In vocal learning species, which rely more heavily on vocalizations for communication and affiliation, turn-taking is likely essential for forming and maintaining social bonds. In this manuscript Benichov and Vallentin provide new insights into how neural circuits control the timing of vocalizations during vocal-turn taking. They show that inhibition of HVC disrupts precision of vocal turn-taking. Using intracellular recordings from premotor and interneurons in HVC they provide evidence that inhibitory circuits play a role in call timing and then confirm this prediction using gabazine application.

Overall, the experimental approach and results are compelling. The conclusions are well supported by the experimental evidence provided. This research is well suited for publication in Nature Communications and the authors should be commended on a nice set of experiments that help advance our understanding of this important scientific problem.

My central, but addressable, concern is that there is a lack of detail for many of the experiments. As written, this leaves several questions about whether proper experimental controls have been done and if other labs could reproduce the present results. I believe that the authors should be able to address the concerns enumerated below by expanding their methods and results sections. It is unlikely that new experiments will be needed to fully address my concerns.

1. For the muscimol and gabazine experiments the authors use gel foam sponges applied to the surface of the brain. HVC is located on the floor of the lateral telencephalic ventricle, not at the brain surface, therefore it is important to provide further details regarding the effectiveness of this strategy. Is the dura still intact? How effectively can drugs spread across the ventricle and into HVC? How far into HVC do pharmacological agents spread and how long does it take for this spread? The medial and lateral portions of HVC are different depths from the brain surface and the thickness of the APH overlying HVC can vary greatly across medial-lateral, anterior-posterior boundaries. Have the authors quantitated the effectiveness of their approach? Can they supply images using fluorescently tagged muscimol or verify effects using reverse microdialysis probes inserted to HVC.

2. For intracellular experiments, I presume the authors are recording from HVC-RA neurons and interneurons. In the results they mention using antidromic activation to identify "premotor neurons", but it is not clear in the results what type of premotor neurons are being recorded from and if all (or what percentage) are identified by antidromic activation or some other aspect of their activity. HVC-X neurons could also be considered premotor. It should be clarified which exact population of neurons is being recorded from and how the authors are verifying this in the results and methods. The same holds true for the putative interneuron recordings. It is stated that in the methods that interneurons are characterized by their high firing characteristics or by their high spike jitter after antidromic stimulations – this is somewhat vague. How many neurons were characterized by each or both methods. Were any cells filled to independently confirm? I understand that the firing characteristics of HVC neuron subtypes are well described, but there is variability among some HVC-RA neurons and some interneurons and more descriptive information about which cells were recorded from and how the authors verify their recordings would be useful throughout.

3. Additional justification for the markov model, which is based on singly housed birds, is needed. Seems like models based on birds in triads would be useful for making predictions for how singly housed birds respond to jamming signals?

Typographical and other minor issues:

Line 54: A brief mention of what "tet" and "stack" calls are would be helpful for non-songbird readers.

Line 58: ..."flexibly"

Line 61: Which subcortical area? If not specifically the name of the area, which part of the brain?

Line 84: extra 'and'

Lines 93 – 97: Was the alternating pair excluded from the investigation into specific neural mechanisms? If not, did it display similar or different responses to the various pharmacological manipulations?

Lines 102-103: Not sure 'catch cycles' are enough to distinguish between these two possibilities. What about putting the later calling bird, 'bird 2', back into the solo trial, to see if it gradually shifts its call timing earlier.

Line 137: Make clear that these are HVCRA neurons, which are antidromically identified.

Line 145: should state action potentials "at and before onset"

Line 185: extra 's' in 'these'

Line 191: "more temporally stereotyped" — vague, what does that refer to? less variability in timing?

Line 228: delete extra 'this'

Line 237: 'occurs'

Line 427: delete 'to'

Review by Todd Roberts

Reviewer #2 (Remarks to the Author):

Summary: The authors explore the behavioral and neurophysiological nature of vocal turn-taking in adult male zebra finches. They focus on song nucleus HVC and demonstrate its critical role in song timing during social interactions particularly the role of local inhibition using a sophisticated multi-prong experimental design. The manuscript is well-written, the figures clear and convincing and the results are fascinating. The statistical tests used are appropriate. I have only minor comments/revisions to consider.

Introduction: This section is well-written and to the point, bringing in appropriate background literature.

1) minor grammatical issue- change 'flexible' to 'flexibility' in line 58.

2) minor grammatical issue- remove the word 'and' in line 84

Results:

- 1) How many practice sessions were needed to coordinate vocal timing to avoid overlap? I assume that the bird pairs needed to become familiar with each other first as well as the playback sound?
- 2) The use of a 'stack' call for playback is mentioned and in the Supplements, it is referenced as a harmonic syllable. I recommend using this word 'harmonic' here in the main text since this word is commonly used in the field when measuring pitch.
- 3) Is there precedence for the use of the Monte Carlo method (line 108)? If so, relevant literature should be cited. Otherwise, the authors should take credit for the first application of this method to song data and its availability to be used by others in the field.
- 4) Because the only spectral measurements made were of pitch and goodness of pitch in Fig. 1, I would caution the authors about making the following statement below in line 115. If you have Wiener Entropy data to add to this figure (as done in Fig. 2), it would further support your statement.
Zebra finches neither altered the duration nor the spectral features of calls across contexts (Fig. 1j; 116 Supplementary Fig. 1c, d) indicating that vocal timing can be controlled independently from 117 acoustic structure during interactions.
- 5) Figure 1, line 427, remove the extra word 'to'
Fig. 1, lines 428-what is the interpretation of the difference in CV scores between the playback alone condition vs. the triad condition? Is it because each bird has to adjust its vocal turn taking to two other competing sounds versus just one, the playback condition or, is it harder for a bird to adjust its vocal turn taking with a live bird vs. an artificial recording? Is the call pattern of the artificial recording more stable and therefore easier to anticipate? This can be commented on in the results describing this figure.
- 6) Supplemental Fig. 1c on the mean pitch for calls alone with playback vs. in triad: how many pitch measurements were made per bird? Studies in zebra finch have shown that 20-25 calls are sufficient to detect experimental differences so it is important to note that you have sufficient power in your call data to confirm a lack of significance between the conditions.
- 7) Briefly, reiterate what a 'tet' versus 'stack' call is and the significance of 'te' to vocal turn taking (line 138) with reference to Fig. 3. Are 'tet' calls seen in the song structure? If so, it would be helpful to point them out for the reader.
- 8) Lines 192-194 and Fig. 5: Fig. 5g suggests that gabazine infused in HVC bilaterally leads to more variation in goodness of pitch (is one bird represented in 5g or multiple birds). However, Fig. 5h-i show no significant p-values between the control and gabazine conditions for variance in pitch and duration measures. Given this, how do you reconcile the statement that: "This manipulation also increased the variability of the acoustic
193 structure of calls for four out five birds (Fig. 5f-i; Supplementary Fig. 4) potentially due to an
194 increase in number of active premotor neurons."

Discussion

- 1) Lines 221-222: It is mentioned that finches living in large groups will vocalize louder than their neighbors. Did you measure the amplitude of the calls – were the pairs adjusting their loudness in addition to adjusting song timing?
- 2) Line 226 mentions that practice sessions were not needed before the finches adjusted their vocal turn taking. Were the pairs familiar to each other or related? Would this make a difference in their ability to adjust quickly?
- 3) The study is based on pairs of male finches so I assume the vocal turn taking is important for territorial defense or between father and son? Do you expect the same mechanisms to be engaged

when the male finch is vocal turn-taking with female finches as potential mates? Given that female finches lack a well-developed HVC song nucleus, do they interrupt the males during their courtship song and are unable to wait their turn? Is there any literature on this?

Julie E. Miller, Ph.D.

Assistant Professor, University of Arizona

Reviewers' comments:

We would like to thank both reviewers for their helpful comments and suggestions which led us to substantially extend our method section and, we think, improve our manuscript. Below we provide detailed responses (in blue) to the reviewers' comments (in black).

Reviewer #1 (Remarks to the Author):

Vocal interactions among individuals involve vocal turn-taking. This type of social interaction is common even among animals that do not learn their vocalizations. In vocal learning species, which rely more heavily on vocalizations for communication and affiliation, turn-taking is likely essential for forming and maintaining social bonds. In this manuscript Benichov and Vallentin provide new insights into how neural circuits control the timing of vocalizations during vocal-turn taking. They show that inhibition of HVC disrupts precision of vocal turn-taking. Using intracellular recordings from premotor and interneurons in HVC they provide evidence that inhibitory circuits play a role in call timing and then confirm this prediction using gabazine application.

Overall, the experimental approach and results are compelling. The conclusions are well supported by the experimental evidence provided. This research is well suited for publication in Nature Communications and the authors should be commended on a nice set of experiments that help advance our understanding of this important scientific problem.

My central, but addressable, concern is that there is a lack of detail for many of the experiments. As written, this leaves several questions about whether proper experimental controls have been done and if other labs could reproduce the present results. I believe that the authors should be able to address the concerns enumerated below by expanding their methods and results sections. It is unlikely that new experiments will be needed to fully address my concerns.

We thank reviewer 1 for the positive assessment and recognize his concern. We have elaborated our methods and results sections to provide further details and clarity about our experimental protocols and controls.

1. For the muscimol and gabazine experiments the authors use gel foam sponges applied to the surface of the brain. HVC is located on the floor of the lateral telencephalic ventricle, not at the brain surface, therefore it is important to provide further details regarding the effectiveness of this strategy.

We agree with the reviewer that we need to provide further details about the pharmacological experiments. In the following we address his concerns point by point:

-Is the dura still intact?

Dura was carefully removed with a tungsten pick after each craniotomy was cut. Before every experimental recording session, the craniotomies were cleaned in order to remove overlying tissue (Supplementary Figure 2a).

Supplementary Material

Lines 90-92:

Due to the presence of APH (area parahippocampalis) above HVC, a thin layer of this tissue (~10-100µm thick) was resected along with dura mater using a fine tungsten pick, directly exposing the central dorsal portion of HVC.

Lines 97-98:

Before and after all experiments, craniotomies were cleaned of any overlying tissue and flushed with saline and fresh silicone elastomer was applied to seal the craniotomies.

-How effectively can drugs spread across the ventricle and into HVC?

We now elaborate on the procedure of how we resected portions of APH with a dura pick to ensure unobstructed access to the surface of HVC (lines 90-92 in Supp. Mat. as above). To prevent the drug from leaking into the adjacent lateral telencephalic ventricle we used a silicone elastomer to seal off the craniotomy borders. This procedure is now illustrated in Supplementary Figure 2a, enabling reproducibility by other labs.

-How far into HVC do pharmacological agents spread and how long does it take for this spread?

The pharmacological agent muscimol has been shown to spread 0.5 -1mm into brain tissue with an asymmetrical diffusion gradient¹. For a more detailed description of the spreading characteristics of muscimol we now cite a systematic study of these dynamics:

Supplementary Material

Lines 88 -90:

Muscimol infusions have been shown to diffuse 0.5-1.0mm through cortical tissue (approximately corresponding to the maximum depth of HVC), with immediate suppression of excitatory transmission upon contact with 10 μ M solution⁷.

Furthermore, we examine the effectiveness of the surface infusion by using electrophysiological recordings of neurons before and after drug application. To assess the spatial extent of gabazine efficacy, we recorded HVC neurons in different recording depths (180-580 μ m) before and after surface-delivered gabazine (lines 103-106 in Supp. Mat. and below). We found that gabazine had the potential to affect the neurons in all depths and that the effect occurred on a short time scale (see Figure 1 below). Please note that we started our behavioral observation experiments after 10 min (line 95 in Supp. Mat.)

Figure 1

Figure 1: HVC-RA neuron recorded before and after surface infusion of gabazine over repeated renditions of song. (Top): Sonogram of bird's song (Bottom): Intracellular recording traces at different time points before and after drug infusion.

-The medial and lateral portions of HVC are different depths from the brain surface and the thickness of the APH overlying HVC can vary greatly across medial-lateral, anterior-posterior boundaries. Have the authors quantitated the effectiveness of their approach?

As stated above and now in lines 90-92 (in Supp. Mat.), APH was removed above HVC. The result from our control experiment with saline demonstrate that the observed behavioral effects are not APH dependent.

-Can they supply images using fluorescently tagged muscimol or verify effects using reverse microdialysis probes inserted to HVC.

Although we did not image fluorescently tagged pharmacological agents or employ microdialysis probes, we now explicitly state that we used electrophysiological recordings to verify the effects of this method of pharmacological infusions.

Supplementary Material:

Lines 103-106:

Efficacy of surface infusion was confirmed with electrophysiological recordings^{8,9} in which the effects of gabazine (increased number of spikes per burst and higher amplitude subthreshold activity) were observed in cells at depths down to 580 μ m from the HVC surface.

With these recordings, we also determined the time course of action to begin within less than 10 minutes of surface infusion. Previously, electrophysiological recordings have been used to verify the effects of pharmacological infusion of bicuculline, muscimol, and gabazine^{2,3} (line 104 in Supp. Mat.).

2. For intracellular experiments, I presume the authors are recording from HVC-RA neurons and interneurons. In the results they mention using antidromic activation to identify "premotor neurons", but it is not clear in the results what type of premotor neurons are being recorded from and if all (or what percentage) are identified by antidromic activation or some other aspect of their activity. HVC-X neurons could also be considered premotor. It should be clarified which exact population of neurons is being recorded from and how the authors are verifying this in the results and methods.

We are thankful to the reviewer for pointing out the need for specificity here. We now clarify that we solely included HVC-RA projecting cells since they are the direct output neurons of HVC to the motor pathway (via RA) (line 139 in Main Text). A detailed description of the antidromic identification protocol was added

Supplementary Material

Lines 118-122:

In order to identify cell types, we stimulated RA with single biphasic (200 μ s per phase) current pulses of < 250 μ A. HVC-RA-projecting premotor neurons were identified by responding to each pulse with a reliable spike with minimal latency jitter (< 50 μ s)¹⁰. For those cells recorded during singing, HVC-RA neurons were further confirmed by the following criteria: 1) song-related depolarization 2) 0-2 bursts of action potentials per motif 3) highly stereotyped subthreshold activity across song repetitions^{11, 12}

-The same holds true for the putative interneuron recordings. It is stated that in the methods that interneurons are characterized by their high firing characteristics or by their high spike jitter after antidromic stimulations – this is somewhat vague.

In the method section we now elaborate on the spiking profile of HVC interneurons and our criteria for classification (lines 123-128 in Supp. Mat. and below).

-How many neurons were characterized by each or both methods.

All HVC-RA neurons were identified based on antidromic identification. As mentioned above, all interneurons had to fulfill strict criteria to be classified as interneurons. These criteria have been clarified in the method section.

Supplementary Material

Lines 118-128:

Interneurons were identified when they fulfilled at least 2 of the following criteria :1) Depolarization during song, 2) displaying phasic spiking activity which is interrupted by short silent gaps and is stereotyped across song repetitions (Note: 6 of 7 interneurons were also recorded during singing and displayed the structured firing, with local minima in their spike rates^{6, 13}), 3) by their high firing characteristics and their high spike time jitter after antidromic stimulation, which was often accompanied by multiple spikes¹⁰, 4) undershooting action potentials below resting membrane potential.

-Were any cells filled to independently confirm?

Although we did not fill any cells for histological analysis, any cells that could not be confirmed via the electrophysiological methods above were not included for further analysis.

-I understand that the firing characteristics of HVC neurons subtypes are well described, but there is variability among some HVC-RA neurons and some interneurons and more descriptive information about which cells were recorded from and how the authors verify their recordings would be useful throughout.

We now provide descriptive information about cell identification (lines 118-128 in Supp. Mat. as above). Unidentified neurons were excluded.

3. Additional justification for the markov model, which is based on singly housed birds, is needed.

In comparing the modeled data from singly housed birds to those birds in triads, we are exploring how much we could expect birds to overlap with each other, if they would not change their behavior. We have rewritten the justification for this simulation to emphasize our goal of assessing behavioral flexibility across contexts

Main Text

Lines 108-112:

To determine if birds in a triad overlap as often as expected if they maintained their response characteristics displayed while alone with playbacks (i.e. no behavioral flexibility), we performed a Monte Carlo simulation of the occurrences of call overlaps of latency-matched birds in the triad context. For this simulation we used each bird's response rates and latency distributions, extracted from the alone with playback condition, as priors.

-Seems like models based on birds in triads would be useful for making predictions for how singly housed birds respond to jamming signals?

We agree that this experiment could help to relate performance in a group setting to each individual's behavior. Unfortunately, we did not directly test the same birds within the triad and with jamming signals while housed singly. Therefore, any predictions made based on the triad data would not be able to be compared to experimental data.

Typographical and other minor issues:

Line 54: A brief mention of what “tet” and “stack” calls are would be helpful for non-songbird readers.

Tets and stacks are now described in more detail.

Main Text

Lines 53-55:

Zebra finches, for instance, exchange thousands of ~50-100ms long, flat harmonic “stack” calls and slightly frequency-modulated harmonic “tet” calls per day¹⁴.

Line 58: ...“flexibly”

Corrected.

Line 61: Which subcortical area? If not specifically the name of the area, which part of the brain?

The subcortical area is now specified as: “the midbrain area known as the dorsomedial part of the intercollicular nucleus (DM)”.

Line 84: extra ‘and’

Corrected.

Lines 93 – 97: Was the alternating pair excluded from the investigation into specific neural mechanisms? If not, did it display similar or different responses to the various pharmacological manipulations?

None of the birds from the pairing experiments underwent pharmacological manipulations.

Lines 102-103: Not sure ‘catch cycles’ are enough to distinguish between these two possibilities. What about putting the later calling bird, ‘bird 2’, back into the solo trial, to see if it gradually shifts its call timing earlier.

We have softened our language in this section by replacing “to determine” with “to examine...”. While not definitive on their own, we believe that differences in response times between catch cycles and solo response are indicative of an ability to adjust timing, and this comparison reduces the possibility of confounding effects of an additional bird’s call biasing response time to a particular call playback.

Line 137: Make clear that these are HVCRA neurons, which are antidromically identified.

Corrected.

Line 145: should state action potentials “at and before onset”

Corrected.

Line 185: extra ‘s’ in ‘these’

Corrected.

Line 191: “more temporally stereotyped” — vague, what does that refer to? less variability in timing?

Changed to “less variable call response timing”

Line 228: delete extra ‘this’

Corrected.

Line 237: ‘occurs’

Corrected.

Line 427: delete ‘to’

Corrected.

Review by Todd Roberts

Reviewer #2 (Remarks to the Author):

Summary: The authors explore the behavioral and neurophysiological nature of vocal turn-taking in adult male zebra finches. They focus on song nucleus HVC and demonstrate its critical role in song timing during social interactions particularly the role of local inhibition using a sophisticated multi-prong experimental design. The manuscript is well-written, the figures clear and convincing and the results are fascinating. The statistical tests used are appropriate. I have only minor comments/revisions to consider.

We thank the reviewer for her appreciation of our work and address her comments in detail below:

Introduction: This section is well-written and to the point, bringing in appropriate background literature.

1) minor grammatical issue- change 'flexible' to 'flexibility' in line 58.

Corrected. Changed to "flexibly" at suggestion of reviewer 1.

2) minor grammatical issue- remove the word 'and' in line 84

Corrected.

Results:

1) How many practice sessions were needed to coordinate vocal timing to avoid overlap? I assume that the bird pairs needed to become familiar with each other first as well as the playback sound?

We thank the reviewer for this question. We now clarify that no practice sessions were needed. Pairs of birds may have been familiar due to the fact that all birds had been housed in a common aviary prior to experiments. Prior to the triad condition, birds had already undergone the 'alone with playback' condition in order for us to characterize their response times to playback. Birds that did not reliably respond to playbacks were excluded from the experiment. We extended our method section to include this information.

Supplementary Material

Lines 45-47:

Prior to testing, all birds had been housed in a common aviary. Birds were first placed in the testing sound box individually and presented with call playbacks. Birds that did not readily and reliably respond to playbacks were excluded from the experiment.

2) The use of a 'stack' call for playback is mentioned and in the Supplements, it is referenced as a harmonic syllable. I recommend using this word 'harmonic' here in the main text since this word is commonly used in the field when measuring pitch.

The harmonic structure of Tet and Stack calls is now described in more detail (lines 54-55 in Main Text)

3) Is there precedence for the use of the Monte Carlo method (line 108)? If so, relevant literature should be cited. Otherwise, the authors should take credit for the first application of this method to song data and its availability to be used by others in the field.

To our knowledge, this exact method has not been implemented within the context of vocal coordination. We will make the code for this analysis available for download.

4) Because the only spectral measurements made were of pitch and goodness of pitch in Fig. 1, I would caution the authors about making the following statement below in line 115. If you have Wiener Entropy data to add to this figure (as done in Fig. 2), it would further support your statement.

Zebra finches neither altered the duration nor the spectral features of calls across contexts (Fig. 1j; 116 Supplementary Fig. 1c, d) indicating that vocal timing can be controlled independently from 117 acoustic structure during interactions.

We thank the reviewer for detecting this omission and we have added results of call entropy analysis to supplemental figures (Supplementary Fig. 1e).

5) Figure 1, line 427, remove the extra word 'to'

Corrected.

Fig. 1, lines 428-what is the interpretation of the difference in CV scores between the playback alone condition vs. the triad condition? Is it because each bird has to adjust its vocal turn taking to two other competing sounds versus just one, the playback condition or, is it harder for a bird to adjust its vocal turn taking with a live bird vs.

an artificial recording? Is the call pattern of the artificial recording more stable and therefore easier to anticipate? This can be commented on in the results describing this figure.

We now specify that the CV score relates to the spread of response latencies across birds and reflects the divergence of response times to the playback when birds are in a triad condition (Figure Legend 1f, line 453-454). We agree with the reviewer that it is an easier task to respond to a perfectly predictable partner (call playback). We now state this in the text (lines 84 and 87 in Main Text).

6) Supplemental Fig. 1c on the mean pitch for calls alone with playback vs. in triad: how many pitch measurements were made per bird? Studies in zebra finch have shown that 20-25 calls are sufficient to detect experimental differences so it is important to note that you have sufficient power in your call data to confirm a lack of significance between the conditions.

The minimum number of calls used to assess the average for each acoustic feature was 54 calls (maximum of 243). This is now stated in the figure legend for Supplementary Fig. 1. All other reported means were based on at least 26 calls.

7) Briefly, reiterate what a 'tet' versus 'stack' call is and the significance of 'te' to vocal turn taking (line 138) with reference to Fig. 3.

We now refer to these calls as "short and acoustically simple" and explicitly connect the functional significance of call related premotor activity to their usage in turn-taking (Lines 142, 144-145 in Main Text)

Are 'tet' calls seen in the song structure? If so, it would be helpful to point them out for the reader.

Tet and stack calls can be similar to song syllables in terms of acoustic structure, and call-like elements can be incorporated into a motif during song development⁴. We now mention this relationship in the main text.

Lines 146-148:

Because of HVC's critical role in song production^{23, 28} and the possibility of "tet"- and "stack"-like elements being incorporated into songs during vocal development²⁹, we tested whether individual premotor neurons can be involved in both song and call production.

8) Lines 192-194 and Fig. 5: Fig. 5g suggests that gabazine infused in HVC bilaterally leads to more variation in goodness of pitch (is one bird represented in 5g or multiple birds). However, Fig. 5h-i show no significant p-values between the control and gabazine conditions for variance in pitch and duration measures. Given this, how do you reconcile the statement that: "This manipulation also increased the variability of the acoustic structure of calls for four out five birds (Fig. 5f-i; Supplementary Fig. 4) potentially due to an increase in number of active premotor neurons."

The scatterplot in 5g is from a single bird (same as in 5f). Although across birds, their average call pitch and durations did not vary, there was an effect of gabazine when the combination of goodness of pitch and pitch are taken into account, across all birds (2-d Kolmogorov-Smirnov test in Supplementary Figure 4). This may have been driven by their individual variability, as the effect was not consistent for a given features across birds.

Discussion

1) Lines 221-222: It is mentioned that finches living in large groups will vocalize louder than their neighbors. Did you measure the amplitude of the calls – were the pairs adjusting their loudness in addition to adjusting song timing?

We thank the reviewer for raising this important point. Due to the size of our experimental cages and resulting variability in the microphone-to-bird distance with movement, we were not confident in the measurements of call amplitude and chose not to focus on loudness here. It has previously been demonstrated, however, that male and female zebra finches will produce louder calls to overcome acoustic masking⁵. We have generalized our statement to "...in a noisy environment." in line 227 and added a reference to this study to more directly support the claim. We also removed the adverb "However" in the following sentence so as to not imply that amplitude changes and timing changes are two mutually exclusive strategies for effective signal transmission.

2) Line 226 mentions that practice sessions were not needed before the finches adjusted their vocal turn taking. Were the pairs familiar to each other or related? Would this make a difference in their ability to adjust quickly?

Because birds were housed in an aviary with 20+ other birds prior to testing, we had not specifically tracked their history of interactions. Since our initial submission, Prior et al.⁶ have reported that the degree of call coordination can vary based on social factors like familiarity. We have added a citation of this work in our discussion (line 233).

3) The study is based on pairs of male finches so I assume the vocal turn taking is important for territorial defense or between father and son? Do you expect the same mechanisms to be engaged when the male finch is vocal turn-taking with female finches as potential mates? Given that female finches lack a well-developed HVC song nucleus, do they interrupt the males during their courtship song and are unable to wait their turn? Is there any literature on this?

Variations in behavior based on pair composition have been explored by D'amelio et al. ⁷ and the recent paper by Prior et al. ⁶ (line 233). We have previously published results showing that female zebra finches are often better than males at adjusting their call timing, and that lesioning the smaller female nucleus RA disrupts their call timing as well. This finding, along with a recent vocal motor pathway tracing study in females ⁸, suggests that the female vocal motor pathway may be more functionally specialized than previously assumed. We have added a brief discussion of female calling in the main text. As for female calling during song, we now cite a review by Heather Williams ⁹, in which females interrupt males during courtship song, with possible function of providing specific feedback.

Main Text

Line 239-241:

This may also be the case in female zebra finches, who do not sing, but can actively coordinate their calls and may rely on a reduced form of the vocal motor pathway to control their timing^{40,43,44,45}.

Julie E. Miller, Ph.D.

Assistant Professor, University of Arizona

References

1. Allen TA, Narayanan NS, Kholodar-Smith DB, Zhao Y, Laubach M, Brown TH. Imaging the spread of reversible brain inactivations using fluorescent muscimol. *J Neurosci Methods* **171**, 30-38 (2008).
2. Hamaguchi K, Mooney R. Recurrent interactions between the input and output of a songbird cortico-basal ganglia pathway are implicated in vocal sequence variability. *J Neurosci* **32**, 11671-11687 (2012).
3. Yanagihara S, Yazaki-Sugiyama Y. Auditory experience-dependent cortical circuit shaping for memory formation in bird song learning. *Nat Commun* **7**, (2016).
4. Lipkind D, Zai AT, Hanuschkin A, Marcus GF, Tchernichovski O, Hahnloser RHR. Songbirds work around computational complexity by learning song vocabulary independently of sequence. *Nat Commun* **8**, (2017).
5. Cynx J, Lewis R, Tavel B, Tse H. Amplitude regulation of vocalizations in noise by a songbird, *Taeniopygia guttata*. *Anim Behav* **56**, 107-113 (1998).
6. Prior NH, Smith E, Dooling RJ, Ball GF. Familiarity enhances moment-to-moment behavioral coordination in zebra finch (*Taeniopygia guttata*) dyads. *J Comp Psychol*, (2019).
7. D'Amelio PB, Trost L, ter Maat A. Vocal exchanges during pair formation and maintenance in the zebra finch (*Taeniopygia guttata*). *Frontiers in Zoology* **14**, (2017).
8. Shaughnessy DW, Hyson RL, Bertram R, Wu W, Johnson F. Female zebra finches do not sing yet share neural pathways necessary for singing in males. *J Comp Neurol* **527**, 843-855 (2019).
9. Williams H. Birdsong and singing behavior. *Ann Ny Acad Sci* **1016**, 1-30 (2004).

REVIEWERS' COMMENTS:

Reviewer #1 (Remarks to the Author):

I have no further comments. Congratulations on a nice manuscript.

Reviewer #2 (Remarks to the Author):

I have reviewed the author's rebuttal, re-submitted main manuscript and supplemental information and have no further requests for clarification. The authors did a great job in responding to reviewers' comments and adding additional helpful details.

Julie Miller, University of Arizona

REVIEWERS' COMMENTS:

Reviewer #1 (Remarks to the Author):

I have no further comments. Congratulations on a nice manuscript.

Reviewer #2 (Remarks to the Author):

I have reviewed the author's rebuttal, re-submitted main manuscript and supplemental information and have no further requests for clarification. The authors did a great job in responding to reviewers' comments and adding additional helpful details.

Julie Miller, University of Arizona

We once again thank both reviewers.